# Panoptic 3D Scene Reconstruction From a Single RGB Image

**Manuel Dahnert**        **Ji Hou**        **Matthias Nießner**        **Angela Dai**

Technical University of Munich

## Abstract

Understanding 3D scenes from a single image is fundamental to a wide variety of tasks, such as for robotics, motion planning, or augmented reality. Existing works in 3D perception from a single RGB image tend to focus on geometric reconstruction only, or geometric reconstruction with semantic segmentation or instance segmentation. Inspired by 2D panoptic segmentation, we propose to unify the tasks of geometric reconstruction, 3D semantic segmentation, and 3D instance segmentation into the task of panoptic 3D scene reconstruction – from a single RGB image, predicting the complete geometric reconstruction of the scene in the camera frustum of the image, along with semantic and instance segmentations. We thus propose a new approach for holistic 3D scene understanding from a single RGB image which learns to lift and propagate 2D features from an input image to a 3D volumetric scene representation. We demonstrate that this holistic view of joint scene reconstruction, semantic, and instance segmentation is beneficial over treating the tasks independently, thus outperforming alternative approaches.

## 1   Introduction

3D scene understanding from a single RGB image is fundamental to many downstream applications, such as for robotics, motion planning, or augmented reality. From a photograph of a scene, one can infer the underlying geometric structures and identify object and structural semantics, even from such a partial, single view observation. Such an understanding is also fundamental towards enabling higher-level interactions with environments, as estimating unseen geometry and semantics without having to observe all parts of an object or scene is paramount for tasks such as robotic interactions, augmented reality, or content creation.

Recently, significant advances have been made towards complete geometric reconstruction from an RGB image input [10, 37], as well as reconstruction of object instances detected in the image [15, 28, 32]; however, 3D semantic reconstruction and instance reconstruction have largely been considered independently. We thus aim to unify the tasks of geometric reconstruction, 3D semantic segmentation, and 3D instance segmentation from a single image, and inspired by the 2D panoptic segmentation task [27], we call this task *panoptic 3D scene reconstruction*. Inspired by the unification of the 2D semantic segmentation and instance segmentation tasks to panoptic segmentation in [27], we similarly adopt a joint semantic segmentation of "stuff" and "things," where "stuff" elements do not have any distinct instances (e.g., structural elements such as walls and floors) and "things" denote semantic objects with instance ids defining the distinct object instances. Thus, for panoptic 3D reconstruction, we aim to predict the surface geometry of the scene from an image, including any missing regions from occlusion, and for each point on the predicted surface geometry, it must be assigned a semantic label and an instance id. To evaluate a predicted panoptic reconstruction, we then evaluate both the segmentation quality and recognition quality of the 3D object instance geometry and geometry corresponding to structural semantic labels.

35th Conference on Neural Information Processing Systems (NeurIPS 2021).

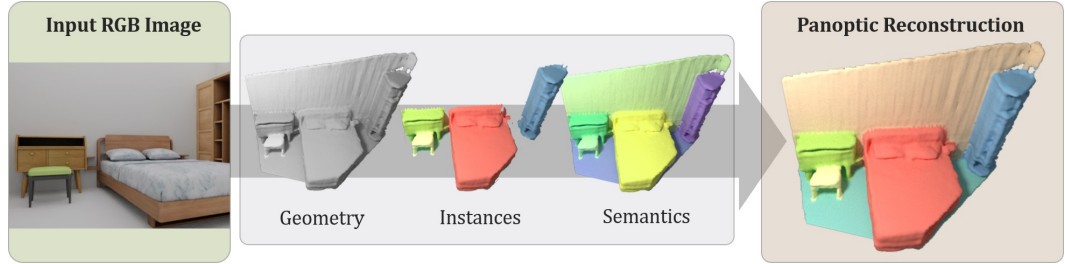

Figure 1: We propose the task of Panoptic 3D Scene Reconstruction from a single RGB image. From an RGB image input, we simultaneously predict geometric reconstruction, semantic labels and object instances, and combine these to form a panoptic reconstruction output.

We believe that this task not only brings together geometric reconstruction, 3D semantic segmentation, and 3D instance segmentation, but also introduces new algorithmic challenges, and hope that the panoptic 3D scene reconstruction task leads to additional insights in holistic 3D perception.

To address the panoptic 3D scene reconstruction task, we propose a new method to jointly reconstruct and segment the observed 3D scene from a RGB image. We first extract features from the image with a 2D convolutional backbone which predicts both depth estimates and 2D instance segmentation. We use the depth estimates to guide the feature lifting to 3D by back-projecting the learned features into a 3D volume of the camera frustum, and develop an instance propagation approach to effectively leverage instance predictions in 2D as seeds for propagation to 3D geometric instances, providing effective object recognition. In contrast to prior approaches for 3D reconstruction from 2D images, we develop a propagation-based approach for object recognition, explicitly bringing 2D predictions to 3D for a refinement rather than direct 3D prediction on a coarse resolution than the 2D signal. We train the object recognition jointly with a reconstruction loss on the predicted scene geometry as well as semantic "stuff" labels, resulting in a holistic 3D semantic scene estimate for an RGB image.

In summary, our contributions are:

- We introduce the task of panoptic 3D scene reconstruction, which aims for holistic 3D scene understanding by jointly predicting scene geometry, semantic labels, and instance ids.

- We propose a new approach for panoptic 3D scene reconstruction by learning to lift 2D features to 3D while propagating learned instance information to 3D object understanding for a robust joint prediction of scene geometry, semantics, and object recognition[1].

- Our method significantly outperforms alternative approaches which treat reconstruction and semantics separately and/or infer only a subset of the outputs.

## 2 Related Work

**Panoptic Segmentation.** Kirillov et al. [27] proposed the task of panoptic segmentation for 2D images, establishing a unified, holistic scene understanding task for image understanding and a panoptic quality metric for evaluation. The task of panoptic segmentation of a single color image is to assign each pixel in the image with an instance label and a semantic label. The semantic label set is composed of "stuff" and "things," where "stuff" labels do not consider instance ids as all pixels belonging to the same "stuff" class belong to the same instance, and "things" labels consider instance ids to distinguish different object instances. This has achieved notable success in inspiring further holistic analysis of image understanding, and we believe such a holistic analysis also plays an important role in 3D perception. We aim to bridge together geometric scene reconstruction [37, 10], 3D semantic segmentation [7, 35], and 3D instance segmentation [40, 20, 42, 12, 26, 17, 41, 29], along with a panoptic 3D scene reconstruction evaluation to capture these characteristics. Narita et al. [31] proposed an approach for joint semantic and instance segmentation on 3D reconstructed scenes, but rely on given 3D scene information without inferring geometry. To the best of our knowledge, our work is the first to bring the panoptic reconstruction task into 3D perception from RGB images.

---

[1]Code can be found at `https://github.com/xheon/panoptic-reconstruction`.

**3D Semantic Scene Completion and Semantic Instance Completion.** Several recent works have explored the task of 3D semantic scene completion, in which both complete scene geometry and semantic segmentation are predicted from partial observations of a scene. From a single RGB-D frame, SSCNet [38] predicts the geometric scene occupancy and semantic class labels in a volumetric grid. [3] follows up on it and proposes to guide the semantic scene completion with coarse 3D sketches of the scene. ScanComplete [8] presented an autoregressive approach for large-scale scan completion and semantic segmentation. Dahnert et al. [6] proposed to retrieve complete objects of partially scanned objects from a shape database by constructing a joint embedding space of scan and CAD objects. Hou et al. [21] and Nie et al. [33] then focused on joint scan completion with instance segmentation of the predicted complete geometry. These approaches have shown the synergies in joint geometric completion along with semantic segmentation or instance segmentation, and we aim to unify all these tasks together for a holistic approach to 3D perception from a single RGB image input.

**Single-View 3D Reconstruction.** Estimating object or scene geometry from only a single RGB image view has been long-studied, with recent advances in deep learning showing significant potential in the reconstruction of shapes from single views [4, 16, 39, 30]. As intermediate representation Huang et al. extend 2D bounding boxes to 3D via joint estimation [23] or perspective constraints [25]. Huang et al. [24] proposes scene grammar, which allows to retrieve CAD models and estimate scene layout to be optimized to fit a single RGB input image. Shin et al. [37] and Denninger et al. [10] recently proposed methods to estimate the scene geometry within the camera frustum of an image view. Recently, Mesh-RCNN [15] pioneered an approach to extend single object reconstruction to RGB images with multiple objects by detecting and reconstructing the object geometry of each detected object in the image, along with their relative orientations. Kuo et al. [28] similarly predicted the object geometry for detected objects in an RGB image, but by CAD model retrieval rather than a generative mesh model. CoReNet [34] omitted explicit object detection and instead extracted 2D information via physically-based ray-traced skip connection between the image and the 3D volume. The approach reconstructs the shape and semantic class of multiple objects from a single RGB image directly in a 3D volumetric grid. Nie et al. [32] then proposed an approach to detect object poses, their geometry, and cuboid room layouts. Similar to Nie et al., we aim to more comprehensively understand both object and scene structures and semantics, but remove any cuboid-structure assumptions, instead lifting 2D features and instance information to 3D for prediction of both "stuff" and "things," and additionally propose a new holistic evaluation metric to understand the full panoptic 3D scene reconstruction quality.

## 3 Method Overview

In the panoptic 3D scene reconstruction task, a single RGB image is taken as input, from which the scene geometry within the camera frustum is predicted, along with semantic and instance label for each surface location on the predicted geometry. We introduce a new method for this task, extracting strong 2D features from the input RGB image. The features are further projected and propagated to 3D, in which the final geometry, semantic, and instance predictions are predicted.

From an input RGB image, we leverage a state-of-the-art 2D instance segmentation network as a 2D backbone to extract features, with proxy losses for 2D instance segmentation and depth estimation. The predicted depth is then used to estimate a coarse 3D representation by back-projecting the learned 2D features and predictions into a sparse volumetric representation. This coarse 3D estimate is then encoded by 3D sparse convolutions, and a coarse-to-fine approach is used to predict high-resolution output as a sparse volumetric representation, with each voxel containing a distance value representing the surface geometry as a distance field, a semantic label, and an instance id. In particular, we exploit the 2D instance predictions by directly propagating them to 3D to be refined, which provides a robust semantic 3D prediction.

## 4 Hybrid 2D-3D Panoptic Reconstruction

**Learning 2D Priors.** An input RGB image $I$ is first encoded by a 2D backbone $\Theta_i$, realized with a ResNet-18 [18] architecture. To learn effective priors for the panoptic 3D scene reconstruction task, we employ proxy losses on $\Theta_i(I)$ for both 2D instance segmentation and depth estimation. We

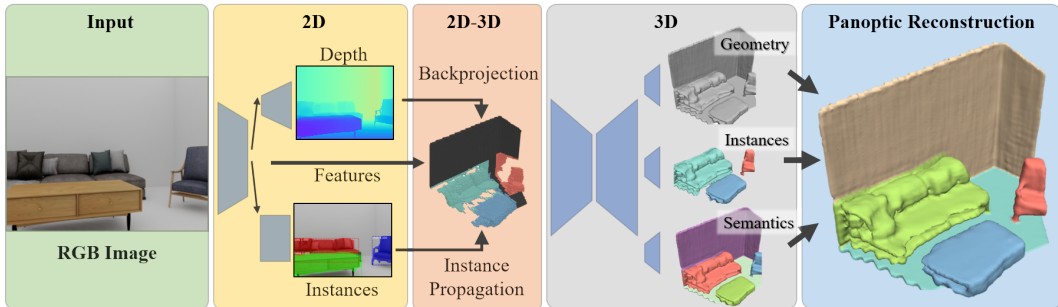

Figure 2: Method overview: from a single RGB image, we extract 2D image features along with depth estimates and 2D instance mask predictions. We back-project both learned features as well as instance predictions to a 3D volume in camera space, thus enabling instance propagation to 3D. The initial 3D estimates are then spatially refined with a sparse 3D generative convolutional network while jointly predicting the scene geometry, semantic labels, and object instance ids within the camera frustum of the image view, including occluded regions. A detailed architecture overview can be found in the supplemental material.

follow Mask R-CNN [19] for the instance segmentation, predicting 2D object bounding boxes, class category labels, and instance masks from $\Theta_i(I)$. For depth estimation, we use the depth estimation decoder of [22], which predicts depth from multiscale features extracted by $\Theta_i$, using losses on the logarithm of the depth errors, gradients, and normals. We then extract features from the output of each ResNet block in $\Theta_i$ as well as the decoder outputs, which are then concatenated to $\theta$ to be projected to 3D.

**Lifting 2D Features to 3D.**  We then lift the 2D features to 3D using the known camera intrinsics of the image and the predicted depth estimates. The 2D features $\theta$, depth estimates, and predicted 2D instance segmentation are back-projected into a $256^3$ 3D volumetric grid using the estimated depth, and encoded as a truncated signed distance field (TSDF) along the view direction, where positive values are in front of the predicted surface, zero encodes the predicted surface, and negative values unseen regions. We use a truncation of $\tau = 3$ voxels, and thus use a sparse 3D representation.

To spatially correlate learned 2D features to directly inform the 3D panoptic reconstruction, the image features $\theta$ as well as instance segmentation mask logits corresponding to each depth estimate copied along the view direction within the truncation, resulting in a sparse 3D feature volume $F = (F_d, F_f, F_i)$ as the concatenation of the computed distance field $F_d$, projected image features $F_f$, and projected instance masks $F_i$. Multiple pixels falling into the same voxel bin are randomly sampled. The 1-D instance segmentation mask logits for each object of a maximum of $N$ detected objects are arbitrarily associated with the channels of an $N + 1$-channel volume, with the extra channel encoding a 1-hot encoding for voxels with no instance mask associations. This enables *instance propagation* from detected 2D object instances to final refined 3D object "things," explicitly leveraging strong 2D priors which result in more robust panoptic reconstruction than relying only on distinction of objects from a much coarser 3D representation than the information from the high-resolution 2D image.

**Sparse Generative Panoptic 3D Scene Reconstruction.**  The panoptic 3D scene reconstruction is then predicted from the feature volume $F = (F_d, F_f, F_i)$ with a sparse generative approach. Each of $(F_d, F_f, F_i)$ are first separately encoded with sparse 3D convolutions to transform the different features characteristics into a common space. The resulting features are then concatenated as $F'$ and encoded together with a sparse 3D encoder using a common sparse encoder $\Theta_s$, realized with a UNet-style architecture [36].

Since the panoptic reconstruction should both refine the estimated depth as well as predict new geometry for unobserved regions, $\Theta_s$ follows the same geometric structure as its input $F'$, spatially downsampling by a factor of 4, while the panoptic reconstruction is predicted in coarse-to-fine fashion, starting with a coarse prediction from $\Theta_S(F')$ using dense 3D convolutions to enable generation of new geometric structures, similar to the sparse generative approach of SG-NN [9] for geometric scene completion. The initial dense prediction is then decoded to sparse panoptic predictions at each spatial hierarchy level, with separate prediction heads for geometric occupancy, semantic labels, and

instance ids, which are then used to inform the next hierarchy level for refined predictions. Similar to [9], we also employ dense and sparse skip connections between the encoder and decoder. At the last hierarchy level, we additionally predicted refined surface geometry as a sparse distance field, along with semantic label and instance ids per voxel.

The final panoptic 3D scene reconstruction is then obtained by extracting surface geometry from the isosurface of the predicted distance field. Each location with distance to the surface less than $\tau_s$ is assigned the semantic label from the 3D semantic segmentation head if that label corresponds with a "stuff" label, and otherwise assigned the semantic label and instance id of the predicted instance mask at the location. Empirically, we found the precedence for "stuff" labels to perform slightly more robustly than the reverse, and provide the nearest semantic and instance labels for any locations that may have not had any semantic or instance predictions.

**Losses.** We train our panoptic 3D scene reconstruction with the following loss:

$$\mathcal{L} = w_d \mathcal{L}_d + w_i \mathcal{L}_i + \sum_h \left( w_g \mathcal{L}_g^h + w_s \mathcal{L}_s^h + w_o \mathcal{L}_o^h \right), \tag{1}$$

where $\mathcal{L}_d$ denotes the 2D depth estimation loss, $\mathcal{L}_i$ the 2D instance segmentation, and at each hierarchy level $h$, a loss for geometry $\mathcal{L}_g^h$, semantic label $\mathcal{L}_s^h$, and instance id $\mathcal{L}_o^h$. Note that the loss is not computed for any regions of the 3D predictions that lie outside of the image view frustum.

For 2D depth estimation loss $\mathcal{L}_d$, we follow the losses used by [22], using a log-encoded $\ell_1$ loss on the predicted depth values along with a cosine normal consistency loss and a gradient loss to regularize the depth estimation. The 2D instance segmentation loss $\mathcal{L}_i$ follows the standard Mask R-CNN [19] procedure, with losses on the regressed 2D boxes, class category labels, and binary masks.

Our panoptic 3D reconstruction is trained to predict geometry, semantic labels, and instance ids for sparse voxels within the truncation region. For the geometric loss $\mathcal{L}_g^h$, each hierarchy level is trained to predict geometry occupancy using a binary cross entropy loss, and the final hierarchy level also employs an $\ell_1$ loss on predicted distance values determining the surface geometry:

$$\mathcal{L}_g^h = \text{BCE}(O_{\text{pred}}^h, O_{\text{gt}}^h) + \mathbb{1}_{h==0} \left\| S_{\text{pred}} - S_{\text{gt}} \right\|_1,$$

where $O$ denote the occupancy predictions and ground truth at the hierarchy level, and for the final hierarchy level $h = 0$, $S$ denotes the distance field representing the surface geometry. The occupancy predictions at each level are used to generate the sparse 3D structure used for the sparse 3D convolutions in the following hierarchy level.

Semantic labels are trained with a class-weighted cross entropy loss $\mathcal{L}_s^h$ for each hierarchy level $h$. 3D instance ids are predicted as binary masks corresponding to the detected objects encoded in the channels of $F_i$. That is, each detected object informs an instance mask refinement and propagation from the initial projected 2D mask logits. The 2D object masks are correlated with ground truth 2D masks when the IoU $> 0.5$. The objects are then arbitrarily ordered to enable a cross entropy loss for the 3D predictions, with the correlated ground truth masks assigned to the arbitrary ordering. A visualization of our instance propagation loss is shown in Figure 3. This provides a strong prior on 3D objects from the 2D predictions, from which only a refinement must be learned in 3D, resulting in more robust and accurate object recognition.

## 4.1 Training

We train our approach for panoptic 3D scene reconstruction on a single RTX 2080 Ti. We first jointly pretrain the 2D encoder, depth estimation and 2D instance predictions with an ADAM optimizer using a batch size of 8 and learning rate 1e-4 for 500k iterations ($\approx 5$ days). The learning rate is decreased by a factor of 10 after 250k and 350k iterations. We then train the 3D sparse generative panoptic reconstruction in coarse-to-fine fashion with a batch size of 1, with the hierarchy levels trained for 10k, 5k and 5k each iterations before the next hierarchy level is added to the training. The full hierarchy is then trained for another 300k iterations.

For training on real-world data, we use a model pre-trained on synthetic data, and fine-tune the 2D predictions for 100k iterations with batch size of 8 and learning rate of of 1e-4 that is decayed after 50k iterations, and fine-tune the 3D hierarchy for 5k iterations for each hierarchy level and additional 185k iterations for the final predictions.

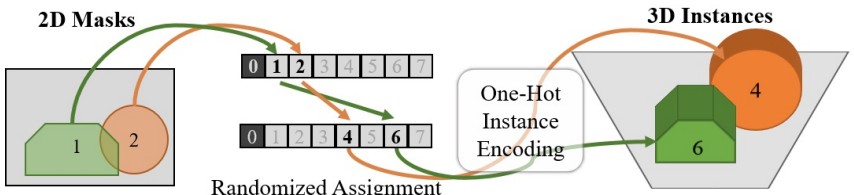

Figure 3: Instance Propagation from 2D to 3D. We directly propagate instance predictions from 2D to 3D, enabling a 3D refinement from high-resolution 2D signal which results in more robust 3D object recognition. During training, each predicted 2D instance mask is matched with a ground truth mask if their IoU$> 0.5$, with matching segments assigned arbitrary instance ids. The ids assigned to predicted 2D masks are then used to assign the mask predictions to the channels of an $N + 1$-channel 3D volume, where $N$ denotes the maximum number of detected objects. and assigned a random instance id which is one-hot encoded and propagated into 3D. This feature volume is then used for refinement in 3D, with the refined output mask logits per channel representing 3D instance masks.

For both datasets, the per-class weights for the class-weighted semantic loss are compute by the inverse log-transformed appearance frequency in the data. Semantic freespace is weighted by 0.001.

## 5 Data

To train and evaluate the task of panoptic 3D scene reconstruction, we consider both synthetic and real-world datasets with dense semantic and instance annotations.

**3D-Front** [14] is a synthetic 3D dataset of indoor scenes, containing 6,801 mid-size apartments with 18,797 rooms populated by 3D shapes from the 3D-Future [13] dataset. We use 11 class categories, 9 of which have instance-level segmentation, and wall and floor as structural "stuff." Excluding empty scenes with no objects, we use a train/val/test split of 4,389/489/1,206 scenes.

To generate image inputs for the panoptic 3D reconstruction task, we render $320 \times 240$ image views of the synthetic 3D rooms. We randomly sample rooms which contain at least one object, and from these rooms, we sample random 10 camera locations per room at 0.75m height above the floor. Views are discarded if they do not contain at least one object in the central region of the image, no objects lie within the 1-7 meter range, or the camera falls within or above an object. We additionally discard views where any rendered 3D objects comprise $< 200$ pixels of the image. We use BlenderProc [11] to generate photo-realistic 2D images along with depth, semantic and instance information. Textures for structural elements were recently added to 3D-Front but not considered here. This results in 96,252/11,204/26,933 train/val/test images.

To generate 3D ground truth, we use SDFGen [1] to generate a 3D volume at 3cm resolution from the same view and cull anything outside of the view frustum. Semantic labels and instance ids are stored for each surface voxel and are inherently consistent with the 2D image views, which were rendered from the 3D data. Note that we only consider the geometry of individual rooms, as predicting geometry or semantics behind walls is highly ambiguous.

**Matterport3D** [2] contains reconstructed RGB-D scans of real-world indoor environments, consisting of 90 buildings reconstructed from RGB-D frames. We use 12 class categories, the same 9 thing classes as with 3D-Front, and wall, floor, and ceiling as stuff classes. We use the official train/val/test split of 61/11/18 scenes.

We use the originally captured RGB frames from the downward and forward looking camera angle as input, and use BlenderProc to render the corresponding 2D semantic, instance, region and depth information for each frame. Image views are discarded from training and evaluation if they contain less than 30% valid depth pixels and the overlap between the visible geometry and complete geometry is at least 20%. 3D supervision is generated by volumetric fusion [5] at 3cm resolution of the depth frames which observe the region within the camera frustum of the input image. In order to avoid fusing ambiguous geometry, e.g. geometry behind walls, we associate each camera location with a Matterport3D region by checking for the intersection between the camera position and the region bounding box. If the camera intersects multiple bounding boxes, the one with the largest overlap with the camera frustum is selected. During training and evaluating we mask depth pixels which belong

to a different region as the camera as these are not present in the ground truth data. This results in 34,737/4,898/8,631 train/val/test images.

# 6 Evaluation and Results

## 6.1 Panoptic Reconstruction Metric

Similar to the 2D panoptic segmentation quality [27], we evaluate panoptic reconstruction quality (PRQ) as an average measure across the class categories, with $\text{PRQ}^c$ for a category $c$:

$$\text{PRQ}^c = \frac{\sum_{(i,j)\in\text{TP}^c} \text{IoU}(i,j)}{|\text{TP}^c| + 0.5|\text{FP}^c| + 0.5|\text{FN}^c|}$$

where TP, FP, and FN denote true positives, false positives, and false negatives for class $c$, respectively, and predicted segments are matched with ground truth via a voxelized IoU overlap of $\geq 25\%$. A segment is considered as the geometry belonging to the same object instance id for things categories, and all surface belonging to a particular stuff category. Note that both predicted and ground truth geometry outside of the camera frustum is not considered in the evaluation, and predicted and ground truth segments are matched by a greedy search for the maximum IoU overlap. All non-matched ground truth segments are considered as false negatives, and all non-matched predicted segments as false positives. The PRQ for a class $c$ can be seen as the product of reconstructed segmentation quality (RSQ) and reconstructed recognition quality (RRQ) for the class $c$:

$$\text{PRQ}^c = \text{RSQ}^c \times \text{RRQ}^c = \left( \frac{\sum_{(i,j)\in\text{TP}} \text{IoU}(i,j)}{|\text{TP}^c|} \right) \times \left( \frac{|\text{TP}^c|}{|\text{TP}^c| + 0.5|\text{FP}^c| + 0.5|\text{FN}^c|} \right).$$

We evaluate panoptic reconstruction by voxelizing predicted panoptic surface meshes at a resolution of 3cm for synthetic data and 6cm for real data. This captures the accuracy of predicted geometric structures as well as semantic classes and instance segmentation. We additionally evaluate these metrics for only things classes and only stuff classes.

## 6.2 Comparison to State of the Art

We compare our approach to state-of-the-art alternatives for the task of panoptic 3D scene reconstruction: SSCNet [38], which predicts geometric occupancy and semantic segmentation of a scene from an RGB-D image; Mesh R-CNN [15], which predicts object instance geometries from an RGB image, and Total3DUnderstanding [32], which predicts object instance geometry and cuboid room layouts from an RGB image. Since SSCNet takes as input an RGB-D image, we provide our predicted depth maps as input along with the RGB image, and obtain additional object instance ids from the semantic labels by applying instance clustering (described in Sec. 6.3). Mesh R-CNN builds from Mask R-CNN [19] to predict object geometry for each detected object, so we evaluate only things classes performance. Total3D similarly predicts object geometry for detected objects in RGB images, with an additional output head to predict the room layout cuboid. We convert Total3D room layout predictions to wall (vertical) and floor (horizontal) predictions for evaluation; note that since Total3D requires known canonical poses for objects in the scenes, which are not provided by the Matterport3D dataset, we do not compare with Total3D on Matterport3D. All evaluated methods have been trained on our generated panoptic data for the respective validation dataset.

Table 1 shows a comparison to these baselines on synthetic 3D-Front [14] data. Our holistic approach, effectively leveraging depth and 2D object recognition for 3D propagation, results in significant improvement in panoptic reconstruction quality, in comparison with state-of-the-art alternatives that tend to predict stuff and things separately, with little to no shared information. As shown in the qualitative comparison in Figure 4, our approach more accurately generates both global structures and object locations as well as local geometric structure, in both observed and occluded regions in the camera frustum.

We also compare with state of the art on real-world image data from Matterport3D [2] in Table 2. The real-world imagery of this data contains much more complexity in scene geometry, object appearance, and arrangement, along with inherent noise in sensor capture, making the panoptic reconstruction task significantly more challenging. Even under this challenging scenario, our holistic approach to panoptic reconstruction results in more effective geometric and semantic understanding than previous approaches which tackled the component tasks independently.

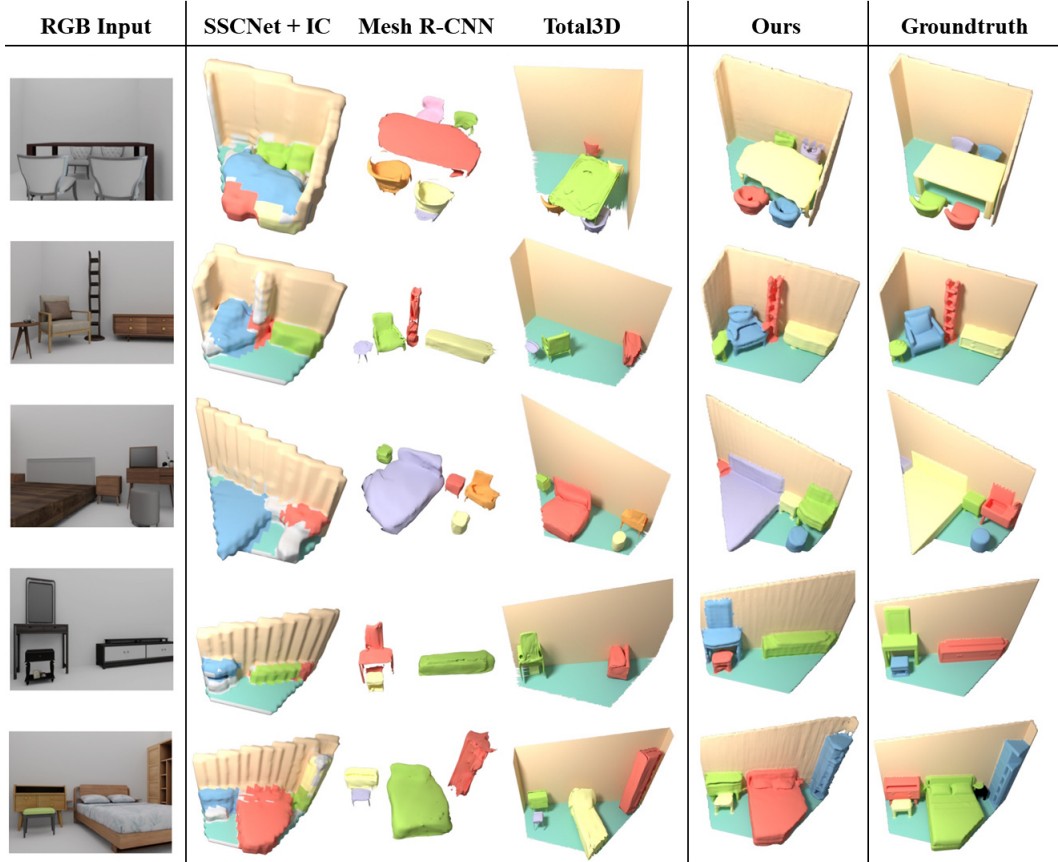

Figure 4: Qualitative comparison of panoptic 3D scene reconstruction on synthetic 3D-Front [14] images, in comparison with SSCNet [38] followed by instance clustering, Mesh R-CNN [15], and Total3D [32]. Our propagation-based joint panoptic reconstruction effectively captures global structures as well as local object arrangements. Different colors denote different instances.

Table 1: Quantitative evaluation of Panoptic Reconstruction Quality on 3D-Front [14].

|  | PRQ | RSQ | RRQ | PRQ | RSQ | RRQ | PRQ | RSQ | RRQ |
|---|---|---|---|---|---|---|---|---|---|
|  |  |  |  |  | *Things* |  |  | *Stuff* |  |
| SSCNet [38] + IC | 11.50 | 32.90 | 33.00 | 8.03 | 32.07 | 24.69 | 26.95 | 36.75 | 70.25 |
| Mesh R-CNN [15] | - | - | - | 20.90 | 38.00 | 53.20 | - | - | - |
| Total3D [32] | 15.08 | 36.63 | 40.15 | 13.77 | 34.88 | 38.89 | 20.94 | 44.49 | 45.85 |
| Ours w/o IP | 20.65 | 53.87 | 29.62 | 8.48 | 48.30 | 15.07 | 75.40 | 78.95 | **95.10** |
| Ours w/o hier. | 44.05 | 55.31 | 70.54 | 37.34 | 50.12 | 65.33 | 74.20 | 78.65 | 93.95 |
| **Ours** | **46.77** | **57.35** | **73.13** | **40.52** | **52.52** | **68.43** | **74.90** | **79.10** | 94.25 |

Table 2: Quantitative evaluation of Panoptic Reconstruction Quality on Matterport3D [2].

|  | PRQ | RSQ | RRQ | PRQ | RSQ | RRQ | PRQ | RSQ | RRQ |
|---|---|---|---|---|---|---|---|---|---|
|  |  |  |  |  | *Things* |  |  | *Stuff* |  |
| SSCNet [38] + IC | 0.49 | 21.68 | 1.50 | 0.19 | 22.75 | 0.59 | 1.43 | 20.43 | 4.43 |
| Mesh R-CNN [15] | - | - | - | 6.29 | **31.12** | 15.60 | - | - | - |
| **Ours** | **7.01** | **28.57** | **17.65** | **6.34** | 26.06 | **16.06** | **10.78** | **40.03** | **26.77** |

| RGB Input | Mesh R-CNN | Ours | Groundtruth |
|:---:|:---:|:---:|:---:|

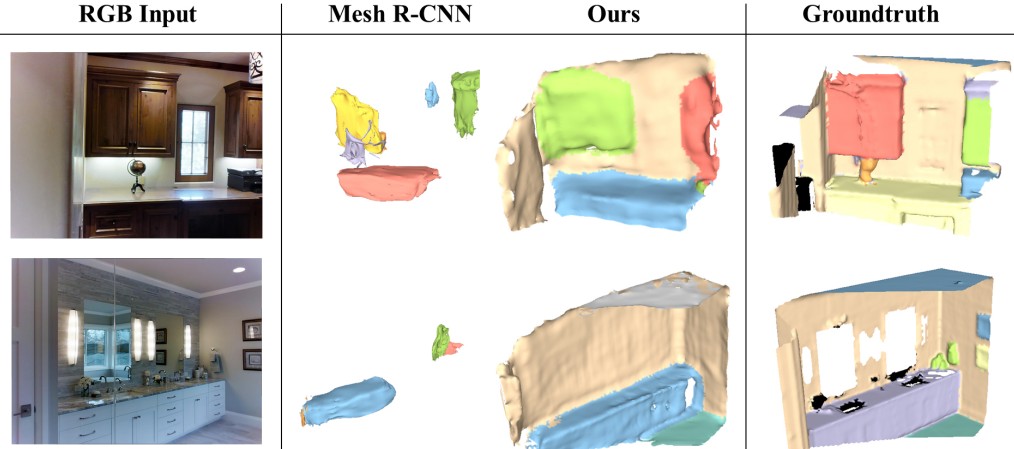

Figure 5: Qualitative comparison of panoptic 3D scene reconstruction from real-world Matterport3D [2] images, comparing the object reconstructions of Mesh R-CNN [15] with our panoptic reconstruction of the scene. Different colors denote different object instances.

## 6.3 Instance Clustering

Instance clustering is a commonly used technique for 3D instance segmentation, used in many state-of-the-art methods [26, 12, 17]. We integrate instance clustering to state-of-the-art comparisons with methods which produce only semantic segmentation, as well as to compare our instance propagation with a 3D instance clustering approach.

We apply instance clustering to semantic segmentation predictions, generating instance predictions in a bottom-up fashion. From the semantic labels, surface locations sharing the same semantic label and lying close together in euclidean space are grouped into instances. A greedy breadth-first search is performed to iteratively find clusters of surface geometry belonging to the same semantic label, where a 3D location is included in the cluster if it is within 2cm of the nearest location in the cluster. A surface location that does not fulfill this requirement then becomes a seed for a new instance cluster. Final output clusters then form instance predictions, where the semantic is given by the semantic label of the locations in the cluster, and the confidence of the instance is the average of the semantic probability scores of the locations in the cluster.

## 6.4 Ablation Studies

**Does instance propagation help?**   In Table 1, we analyze our instance propagation from 2D, in comparison with a 3D instance prediction approach (*Ours w/o IP*), instead using instance clustering as described in Sec. 6.3. Since our instance propagation can directly leverage high-resolution 2D signal to be refined to 3D objects rather than a direct prediction on coarser estimated geometry, our instance propagation results in significantly improved panoptic reconstruction performance.

**What is the effect of the progressive coarse-to-fine prediction?**   We additionally analyze the our coarse-to-fine hierarchical predictions in Table 1, in comparison to an alternative without the use of progressively generated intermediate hierarchy predictions (*Ours w/o hier*). The coarse-to-fine strategy helps to localize predictions early, resulting in a performance improvement.

**What is the effect of providing ground truth depth information?**   We investigate the performance of our approach and baseline methods when we provide ground truth depth information. For our method and SSCNet [38] we train and evaluate with ground truth depth. For MeshR-CNN [15] we feed the ground truth z-location of each object. Here, we also train and compare with Sketch-Aware SSCNet [3]. The results can be found in Table 3.

Table 3: Oracle ablation with providing ground truth depth for SSCNet, Sketch and our approach. For MeshRCNN we supply the ground truth z location of each object.

|  | PRQ *all* | PRQ *Things* | PRQ *Stuff* |
|---|---|---|---|
| SSCNet [38] + IC w/ GT depth | 15.88 | 12.36 | 31.75 |
| Sketch [3] + IC w/ GT depth | 26.12 | 26.23 | 25.60 |
| Mesh R-CNN [15] w/ GT z | - | 35.76 | - |
| **Ours w/ GT depth** | **48.08** | **41.67** | **76.95** |

## 6.5 Limitations

The task of panoptic scene reconstruction sets forth a evaluation of simultaneously accurate geometric reconstruction, semantic labeling, and object recognition. Our propagation-based approach from 2D estimates to 3D refinement shows effective results on synthetic environments, but can sometimes struggle to accurately place things and stuff for real-world observations where there is significantly more noise and clutter, which can result in shifts of predicted structures. We believe that stronger priors on object sizes and relations are a promising avenue, as well as further integration of the joint prediction of stuff and things from 2D to 3D.

## 7 Conclusion

We introduced the task of panoptic 3D scene reconstruction from a single RGB image, to predict a holistic understanding of the image encompassing geometric reconstruction, 3D semantic segmentation, and 3D instance segmentation. Our new method for panoptic reconstruction effectively leverages lifting both learned 2D features as well as predicted depth and 2D instances to 3D, significantly outperforming state-of-the-art alternatives to this task. We hope that the panoptic reconstruction task will help to drive forward research towards more comprehensive holistic scene understanding, of which structural and geometric understanding play a fundamental part.

## Broader Impact

Our work proposes a new, holistic scene understanding task of panoptic 3D scene reconstruction from a single RGB image, which lies at the basis of many visual perception tasks, such as autonomous navigation, mixed reality, or content creation. As our method is heavily data-driven, we must be aware of the use of data and any new data captures to ensure consent and privacy considerations, as well as possible unintended biases from data captured in a limited set of locations.

## Acknowledgements

This project is funded by the Bavarian State Ministry of Science and the Arts and coordinated by the Bavarian Research Institute for Digital Transformation (bidt), the TUM Institute of Advanced Studies (TUM-IAS), the ERC Starting Grant Scan2CAD (804724), and the German Research Foundation (DFG) Grant Making Machine Learning on Static and Dynamic 3D Data Practical.

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
