# OpenReview forum: "Panoptic 3D Scene Reconstruction From a Single RGB Image"
_NeurIPS.cc/2021/Conference — NeurIPS 2021 Poster_

### Official Review · Reviewer_faBK · 2021-07-16

**Rating:** 6
**Confidence:** 4

**Summary:**

The paper proposes an approach for panoptic 3D reconstruction that combines, in an end-to-end trainable manner: (i) 2D depth from rgb, (ii) mask rcnn-style instance segmentation and (iii) a final refinement stage to produce the overall final result.

**Limitations And Societal Impact:**

-

**Main Review:**

Positive:
* While I am somewhat uninformed about the state of the art, this does appear to be the first paper to propose an end to end trained panoptic 3D reconstruction (/ stuff + instance segmentation) solution.
* The paper is well written and easy to understand.

Negative / Questions:
* The key components of the approach are not entirely novel or special, with the various bits of the system being used before in previous tasks (eg the standard-ish depth generation and 2D instance segmentation, traditional-ish lifting, sparse + dense Unet as used before in shape completion).
* The overall structure is also somewhat predictable (except for, maybe, the sparse + dense unet component).
* Literature review seems rather brief, especially in the single view 3D reconstruction section, eg CoReNet and Points2Objects (and related) seem to be missing, though would be quite relevant.
* A version of Figure 1 + 2 from the supplementary should be in the main paper, since it aids quite a bit in understanding the proposed method.
* A benchmark / ablation of sparse vs dense 3D CNNs at the start of the network could be added.
* Given the bottoms-up nature of the approach, a baseline of eg backprojected / lifted 2D panoptic segmentation would seem like a natural approach to investigate / compare against. Is this included in the paper?

Overall, while I do think the solution is rather simple (and reads a bit like a concatenation of previous papers), I am for acceptance because of the novelty of the task, the good results and the quality of the writing.

**Time Spent Reviewing:**

2

---

> ### Author Response · Authors · 2021-08-10
> **Reply to Reviewer faBk**
>
> Thank you for the valuable feedback!
>
> **1. Relation to CoReNet and Points2Objects.**
>
> Thanks for pointing out the missing literature, we will include and add discussions in the final paper.
> Similar to Mesh RCNN, both approaches predict individual objects, but do not reconstruct the scene geometry. We will also include a comparison to CoReNet on the panoptic reconstruction metrics.
>
> **2. Supplementary Figs.1+2 to the main paper.**
>
> Thanks for the suggestion, we will add it to the main paper.
>
> **3. Ablation of sparse vs dense 3D CNNs.**
>
> Thanks for suggesting to compare these different options for the 3D encoder part of our network. We note that for the task of scene reconstruction from a single image, we aim to predict the scene geometry and semantics within the camera frustum (similar to reconstruction approaches from RGB or 2.5D input), for which a sparse representation is well-suited to only explicitly encode surface geometry within the frustum, whereas a dense volumetric grid will encompass regions outside of the camera frustum as well as significant amounts of empty space. Such a dense approach thus limits the spatial resolution or network capacity which fits into memory of current GPUs for training.
> We leverage sparse 3D CNNs to enable predicting higher resolutions within the camera frustum, where the use of a dense 3D CNN did not fit into memory of a RTX 2080 Ti for training at 3cm^3 voxel resolution. In practice, we observed similar performance with sparse and dense CNNs at the same resolution, but are able to achieve much higher output resolutions with sparse 3D CNNs, which is also naturally intuitive as higher resolutions encode significantly more empty space in a dense representation; this leads to an overall better reconstruction performance; we will add this ablation to the final paper.
>
> **4. Lifting 2D panoptic predictions to 3D.**
> We evaluate lifted 2D predictions from Mask R-CNN to 3D (using the predicted depth) in Table 1 of the supplemental material. Since this kind of approach cannot represent any occluded regions or leverage explicit spatial 3D reasoning, it results in noticeably worse performance (8.94 PRQ vs our 40.52 PRQ on things). We believe lifting a full 2D panoptic prediction to 3D by predicted depth would exhibit similar trends.

---

> > ### Comment · Reviewer_faBK · 2021-08-27
> > **Maintaining rating**
> >
> > Thank you for the comments and for the various improvements to the paper. I am happy to maintain my rating.

---

### Official Review · Reviewer_9SvP · 2021-07-16

**Rating:** 6
**Confidence:** 4

**Summary:**

The authors define a new problem called "panoptic 3D scene reconstruction" - a joint task of 3D geometry prediction, semantic segmentation, and instance segmentation, from a single RGB image. To address this new task, a two-stage method is proposed. In the first stage, 2D CNN is used to predict 2D features, a depth image, and 2D instance segmentations. In the second stage, these 2D features and instance segmentation results are back-projected into 3D, and then passed into a 3D CNN for the final 3D predictions. This proposed pipeline is demonstrated on the synthetic 3D-front dataset, as well as the Matterport3D real-world dataset, and significantly outperforms prior related work.

**Limitations And Societal Impact:**

Yes

**Main Review:**

Originality

The newly defined "panoptic 3D scene reconstruction" problem is a natural extension of 2D panoptic segmentation. The components used in the proposed pipeline, including reusing 2D features, uplifting 2D features to 3D via backprojection, multi-task joint training, and 3D CNNs, are all well studied in the community.

Quality

Based on the evaluation metrics, the proposed method significantly outperforms related prior work in terms of geometry, semantic segmentation, and instance segmentation. The example predictions also seem to align better with the ground truth. However, the experimental section is relatively simple, and I do have questions about the proposed method and some suggestions regarding ablation studies.

1. Based on Section 4.1, it is my understanding that the 2D network and the 3D network are trained sequentially and separately, rather than jointly. Could you elaborate on the reasoning behind this design choice? Does joint training lead to sub-optimal results?
2. The current evalutions are all based on 3D volumetric representations, but not 2D. When the 3D predictions are rendered back into the image space, how is the quality of depth, semantic mask, and instance segmentation? Does the 3D network improve the quality, or decrease it?
3. According to Section 4.1, the real-world model is pretrained on synthetic data and then fine-tuned on Matterport3D. Is this the same training procedure for the prior work for a fair comparison? Is there additional effort to make sure that the prior method (e.g. Mesh R-CNN [13]) is trained properly on these datasets and is indeed achieving their optimal performance?
4. The comparison against SSCNet [31] seems a bit strange, given that the method expects a single (good) depth image as input. A more fair experiment would be to use ground truth depth in both SSCNet and the proposed method.
5. After reading the paper, it is still unclear to me why there has been a significant improvement in the metrics compared to prior work - is it because of joint task learning providing additional supervision and task-shared feature representations, or is the proposed network architecture far better than prior related work? Please elaborate and consider adding ablation studies to support your arguments.
6. On Line 127, viewing directions are also used as input - does this help improve quality?

Clarity

The paper is straightforward and easy to follow.

Significance

I believe panoptic 3D scene reconstruction, and more generally multi-task prediction in 3D, is an important topic in the computer vision / machine learning community.

**Time Spent Reviewing:**

3

---

> ### Author Response · Authors · 2021-08-10
> **Reply to Reviewer 9SvP**
>
> We thank the reviewer for the feedback and comments on our work!
>
> **1. Why use sequential training rather than end-to-end training?**
>
> In our experiments, we did not observe a significant improvement with end-to-end training, while sequential training typically resulted in faster training (without the full backpropagation).
>
>
> **2. How is the quality of depth, semantic mask, and instance segmentation when the 3D predictions are rendered back into the image space? Does the 3D network improve the quality, or decrease it?**
>
> We thank the reviewers for proposing this interesting comparison. We will post the results of this evaluation here.
>
>
> **3. Comparisons on Matterport3D (i.e. pre-training on synth, fine tuning real).**
>
> For comparisons on Matterport3D data, we used the pre-trained 3D-Front models to initialize fine-tuning on Matterport3D data to achieve the final results.
>
>
> **4. Comparison to SSCNet with GT depth.**
>
> We are happy to evaluate SSCNet with ground truth depth as well as MeshRCNN with ground truth z for each object, respectively on the 3D-Front data:
>
> |                    | PRQ (all) | PRQ (things) | PRQ (stuff) |
> |--------------------|:---------:|:------------:|:-----------:|
> | SSCNet w/ GT depth |    15.88% |       12.36% |      31.75% |
> | Sketch w/ GT depth |    26.12% |       26.23% |      25.60% |
> | MeshRCNN w/ GT z   |         - |       35.76% |           - |
> | Ours w/ GT depth   |   	48.08%  |       41.67% |      76.95% |
>
> Edit: Add Chen et al. - 3D Sketch-aware Semantic Scene Completion via Semi-supervised Structure Prior results.
>
> **5. Improvements over prior works.**
>
> Compared with existing works, our approach maintains several advantages. For instance, in comparison with object-focused approaches such as Mesh-RCNN which predicts object reconstructions and their 3D centers, our approach estimates the full surface geometry of the scene in absolute scene coordinates, thus enabling more accurate 3D localization and recognition. Our coarse-to-fine sparse generative model additionally allows 3D reconstruction at much higher resolutions (e.g., Mesh R-CNN’s object meshes are constructed from 28^3 voxelizations, limiting the output resolution). Our holistic surface reconstruction along with semantic estimation also enables more robust semantic 3D reconstruction in comparison to methods such as Total3DUnderstanding which predict room layout boxes and objects whose locations are explicitly represented as only a few parameters but can be more strongly impacted by small errors (e.g., converging to mean locations).
>
> **6. View directions (L127).**
> We use knowledge of the camera intrinsics in order to back-project learned 2D features, but do not encode the view directions explicitly into the 3D representation (features are integrated into the 3D volume along the viewing ray, but only 2D features and depth/instance predictions are encoded in 3D).

---

> > ### Comment · Reviewer_9SvP · 2021-08-27
> > **Maintaining rating**
> >
> > Thank you for your response. The additional experiment using ground truth depth is helpful and convincing, and please incorporate these clarifications into the updated version.
> >
> > However I am not completely convinced by authors' argument on why the proposed method outperforms prior work, especially given the lack of more detailed ablation studies. The authors also did not respond to my question on whether additional efforts were made to ensure fair comparison - for instance, as far as I know MeshRCNN was not trained on 3D-Front or Matterport3D, so how did the authors know their training was with the right set of hyper parameters (learning rates, batch sizes, etc.) and was producing the optimal result for the baseline methods? Why not train and test on Pix3D dataset like MeshRCNN?
> >
> > I will maintain my rating at 6 (Marginally above the acceptance threshold).

---

> > > ### Author Response · Authors · 2021-08-28
> > > **Reply to Reviewer 9SvP**
> > >
> > > Thanks for the suggestions - we are happy to include the additional experiments and clarifications to help improve the paper.
> > >
> > > **Fair comparisons:** we spend significant efforts in finding the right hyperparameters to ensure that the baselines were in the correct settings to achieve their best performance. All methods reported in the paper (including MeshRCNN) were trained and evaluated on 3D-Front and Matterport3D. For evaluation on Matterport3D, all methods were fine-tuned on Matterport3D from the corresponding 3D-Front-pre-trained model.
> > >
> > > **Why not evaluate on Pix3D:** The major challenge is that Pix3D has no absolute scale annotations for objects; i.e., all objects have similar scales relative to each other regardless of common metric sizes. However, the absolute estimation of scale/depth is fundamental to our task of Panoptic 3D Scene Reconstruction. In addition, Pix3D is largely focused on scenes with only one object annotation - in contrast, our work focuses on the prediction of the entire scene within the camera frustrum rather than the reconstruction of individual objects. We thus instead evaluate the panoptic 3D scene reconstruction task on 3D-Front and Matterport3D, as both datasets provide all panoptic 3D information.
> > >
> > > **Why does it outperform prior work, e.g., MeshRCNN:** Our method leverages the context of the entire camera frustrum (objects + stuff) which enables learning of larger-scale structures and thus stronger 3D features. In contrast, MeshRCNN is an object-by-object reconstruction method; i.e, the features of each detected object from the MaskRCNN backbone are processed by a separate reconstruction head and after detection, there is no feature interaction between the geometry / 3D semantics of different objects.

---

### Official Review · Reviewer_8qDd · 2021-07-16

**Rating:** 8
**Confidence:** 4

**Summary:**

This paper defines and presents a method for the novel task of panoptic 3D scene reconstruction, i.e. jointly predicting complete 3D scene geometry, 3D semantic and 3D instance labels. The method consists of using backbone 2D models for depth and instance segmentation and lifting them into 3D before processing with a SG-NN [5] like model to complete and refine the initial volume into instance, semantic and occupancy/TSDF volumes. The authors propose an "instance propagation" method to associate 2D instance masks with 3D instances which is shown to improve over a post hoc "instance clustering" in a 3D semantic only volume. Experiments on a synthetic dataset (3D Front) and Matterport3D show improvement over prior works , some of which tackle a subset of the problem (e.g. only scene structure and semantics).

The primary contribution of this paper is the definition of this new holistic panoptic 3D scene reconstruction task and a new method and evaluation metric for the same with improved performance over prior works which tackle these individual tasks in isolation.

**Limitations And Societal Impact:**

I believe the authors have addressed some limitations in Sec 6.5 of the paper. I would appreciate responses to the questions mentioned in the clarifications section above.

**Main Review:**

*Strengths*
- 3D scene structure, 3D semantic and 3D instance segmentation are critical to a holistic understanding of the scene and have been seeing a lot of progress in recent years. Similar to the 2D analog of panoptic segmentation, 3D panoptic reconstruction is the next logical step in this evolution and this paper helps define it with metrics and a method for it. This is quite an important contribution in my opinion.

- The paper is well written and the illustrations make it easier to understand the paper (e.g. Fig 3 helps explain the instance propagation idea well). The visual results are quite impressive.

- The comparisons against prior work (Table 1 and 2) have been carefully done with proper explanation of the alternative used when an existing system wasn't directly applicable in this setting (e.g. input depth for SSCNet, Total3D requiring canonical poses for objects).

*Weaknesses*
- One concern I have with this paper is with technical novelty. At the surface it seems like a simple combination of existing depth + instance segmentation algorithms combined with prior work for scene completion (SG-NN) with extension to semantic and instance label volumes. The instance propagation idea is slightly novel but perhaps not enough. However, defining the 3D panoptic scene reconstruction task and proposing a metric for it could be considered a significant enough contribution.

*Clarifications*
  - L120 - How are multiple feature map from different spatial resolution from the ResNet concatented into $\theta$?
  - Why was the choice to randomly select a pixel for a voxel with multiple backprojections made? Why would you not use standard bilinear sampling for the same?
  - In Table 2, Mesh R-CNN seems to show significantly better performance on "Things" over this method. What is the primary reason for this?



**Time Spent Reviewing:**

2

---

> ### Author Response · Authors · 2021-08-10
> **Reply to Reviewer 8qDd**
>
> Thanks for the review and helpful comments!
>
> **1. Novelty and Contribution.**
>
> Our main contribution is to propose and evaluate the new task of 3D panoptic reconstruction as the next step in 3D perception from a single image. To this end, we introduce an approach to jointly predict geometric reconstruction, 3D semantic segmentation, and 3D instance segmentation from an RGB image leveraging lifting and propagation of 2D features to a 3D representation.
>
> **2. How are multiple feature maps from different spatial resolutions from the ResNet concatenated?**
>
> We already have the feature maps in the same spatial resolution due to the Multi-scale Feature Fusion module (Hu et al. [19]), thus the multiple 2D features maps can be concatenated channel-wise.
>
> **3. Pixel<->Voxel Backprojection.**
>
> We voxelize the back-projected features of 2D pixels by MinkowskiEngine, where the default setup is to randomly select one pixel that falls into the voxel.
>
> **4. Mesh R-CNN performance on things in Tab. 2.**
> While we outperform Mesh R-CNN in panoptic reconstruction quality on things, the reconstructed segmentation quality of Mesh R-CNN remains slightly higher as it has a tendency to predict more objects while our approach slightly underpredicts objects (the tendency to overpredict is penalized more heavily in the full PRQ evaluation).

---

> ### Comment · Reviewer_8qDd · 2021-09-01
> **Maintaining rating after rebuttal**
>
> I would like to thank the authors for the clarifications in the rebuttal. I still believe this is a strong paper and would like to maintain my rating of accepting this paper at NeurIPS.

---

### Official Review · Reviewer_iPdb · 2021-07-18

**Rating:** 6
**Confidence:** 3

**Summary:**

The authors propose a way to perform panoptic 3d scene reconstruction from a single image. Depth and instance segmentation from an rgb image help create a 3d volume through back-projection which is subsequently refined to produce semantics, instances and 3d reconstruction, which effectively translates to a panoptic representation.



**Limitations And Societal Impact:**

The authors discuss limitations and societal impact of the work.

**Main Review:**

Strong Points
- The authors are able to perform the hard task of panoptic reconstruction from a single image
- The method works better than SSCNet and Mesh R-CNN

Weak Points
- The evaluation of the method is severely lacking. As the authors produce 3d reconstruction and semantic segmentation in addition to instance ids, there should be proper evaluation for both these outputs against state of the art.
- I understand that no other method does panoptic 3d reconstruction. However, the authors should 'downgrade' panoptic segmentation and compare with other semantic segmentation methods as opposed to 'upgrading' an existing method to produce panoptic labels. This will avoid bias and validate the usefulness the proposed approach at a more modular level.
- Several relevant approaches have not been discussed and compared against. For e.g., CoReNet by Popov et. al. should be compared for the task of 3d reconstruction and 3D Sketch-aware Semantic Scene Completion via Semi-supervised Structure Prior by Chen et.al. is a better and more recent baseline compared to SSCNet.


**Time Spent Reviewing:**

1.5

---

> ### Author Response · Authors · 2021-08-10
> **Reply to Reviewer iPdb**
>
> Thanks for the comments and suggestions!
>
> **1. Relation to CoReNet and SketchAwareSSC.**
>
> We thank the reviewer for mentioning the additional baselines to help improve our paper. We will include additional discussion of CoReNet and SketchAwareSSC; CoReNet aims to reconstruct the objects seen in a image (`things`, without `stuff`) leveraging ray-traced skip connections, and SketchAwareSSC leverages boundary features as structural priors for the task of semantic segmentation and geometric completion from a 2.5D input. To compare with both methods, we have followed the official code repositories of CoReNet and 3D Sketch-Aware Semantic Scene Completion to process our data in the corresponding formats and are currently training the models; we will post the panoptic reconstruction results here as soon as possible and include these experiments in the final paper. Note that similar to Mesh R-CNN, since CoReNet only reconstructs objects, we can only evaluate it for “Things”. For Sketch-Aware SSC, we use the same instance clustering approach (as also applied to SSCNet) to produce the required instance information in order to evaluate the full panoptic reconstruction output.
>
> **2. “Downgrading” panoptic segmentation.**
>
> Our aim is to holistically address 3D perception from a single RGB image, thus solving for geometric reconstruction, semantic segmentation, and instance segmentation jointly. However, in order to evaluate our task, we additionally evaluate each prediction independently (e.g., stuff, things, panoptic, segmentation vs recognition, etc.). For object reconstruction, we compare favorably against Mesh R-CNN and Total3DUnderstanding, while for “stuff”, we compare against SSCNet. In a sense these evaluations are “downgrading” the panoptic segmentation to its individual components. However, note that no existing method addresses the full panoptic segmentation task.

---

> ### Author Response · Authors · 2021-08-24
> **Reply to iPdb - Additional comparison with Chen et al. - 3D Sketch-Aware Semantic Scene Completion**
>
> Thanks again for the suggestion to compare with Chen et al. - 3D Sketch-aware Semantic Scene Completion via Semi-supervised Structure Prior, which is a more recent SSC baseline compared to the one in our paper. As suggested by Reviewer 9SvP, we train and evaluate Sketch-Aware SSC with ground truth depth (similar to the SSC baseline). The results can be found in the table below. The improved semantic prediction results in better PRQ performance compared to the original SSC (26.12% vs 15.88%), but is still significantly outperformed by our method (48.08% vs 26.12%). Interestingly, the “stuff” classes wrt. SSC drops in performance due to Sketch-Aware SSC’s tendency to predict somewhat thicker geometry in these regions.
>
>
> |                    | PRQ (all) | PRQ (things) | PRQ (stuff) |
> |--------------------|-----------|--------------|-------------|
> | SSCNet w/ GT depth |    15.88% |       12.36% |      31.75% |
> | Sketch w/ GT depth |    26.12% |       26.23% |      25.60% |
> | MeshRCNN w/ GT z   |         - |       35.76% |           - |
> | Ours w/ GT depth   |   	48.08%  |       41.67% |      76.95% |

---

> > ### Comment · Reviewer_iPdb · 2021-08-26
> > **Updating rating based on rebuttal**
> >
> > Thank you for these additional experiments. The rebuttal satisfies most of my concerns with the paper and I am upgrading the rating of the paper accordingly.

---

### Author Response · Authors · 2021-08-10
**General comment**

We would like to thank the reviewers for the constructive and valuable feedback; we are happy that the reviewers found our proposed task of panoptic 3D reconstruction from a single RGB image to address “an important topic” (9SvP) and to present “an important contribution” (8qDd). We are glad that reviewers recognize that our approach is the first to propose and address the challenging task of panoptic 3D reconstruction from a single image (faBK,iPdb), presented in a “well written and easy to understand” fashion (8qDd,faBK), and supported by “quite impressive” results with comparisons “carefully done with proper explanation” (8qDd).

We believe 3D panoptic reconstruction is an important milestone towards enabling 3D perception from a single RGB image, thus facilitating holistic 3D scene understanding and opening up new research avenues across different areas. In contrast to prior works that focus on semantic segmentation or object reconstruction, we propose to effectively leverage and propagate 2D semantic and instance features to holistic 3D understanding of stuff and things.

---

### Decision · Program_Chairs · 2021-09-27

**Decision:**

Accept (Poster)

**Comment:**

The paper proposes a method to learn panoptic 3D scene reconstruction from a single image. Depth and instance segmentation from an RGB image instantiate a 3D feature volume through back-projection which is subsequently refined to produce semantics, instance segmentations and 3D reconstruction. All reviewers acknowledge the good empirical performance of the method and the high  relevance of the problem tackled for the NeurIPS audience.
Concerns were raised regarding comparisons of the paper to previous methods that did not perform jointly the tasks of segmentation and 3D reconstruction, which the rebuttal addressed. Authors are encouraged to include all additional experiments in the final version.